# Good Life without Happiness

**Timo Airaksinen** 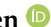

Department of Practical Philosophy, University of Helsinki, FI-00014 Helsinki, Finland;
timo.airaksinen@helsinki.fi

**Abstract:** A good life combines lively living and a good purpose, which depend on action results and consequences. They supervene upon the action results that create life's meaning. A good life is never evil because evil deeds, as such, are not part of the agent's action repertoire. Agents cannot claim them as their own; if they do, dishonest hypocrisy and social stigmatization follow. But, when action results are good, the purpose is good, too. One cannot realize an evil purpose by acting morally. I argue against the idea that a passive, dreaming life could be a good life. I discuss specific kinds of religious life that follow a monastic rule. A good life may not be happy, although it tends to be so. I discuss various theories of happiness, including the traditional Socratic view that virtue and virtue only make an agent happy. I conclude that a good life is not the same as a virtuous life; hence, a good life can be unhappy. To conclude, I discuss personal autonomy in social life. A good life requires that one's actions and goals are one's own, but such ownership is hard to realize because of a social life's complicated and demanding mutual dependencies. I conclude that full ownership is fiction, so a good life is a social life.

**Keywords:** virtue; hedonism; action; purpose; consequence Sisyphus; authority; autonomy

## 1. Meaning of Life and Living

We want a good life; a good life is a meaningful existence with a significant purpose. What does this mean? Life's emptiness indicates its meaninglessness because something essential is lacking. The metaphor of emptiness reveals a paucity of content; hence, an empty life lacks content: it is tedious; boring; dull; repetitive; predictable; and, as such, disappointing. Such a life carries no message: it cannot tell much about a person and their social existence; in other words, one cannot narrate such a life as an individual existence. The person is alive, but he is not living a life. A meaningful existence has enough content to generate a significant narrative.[1]

Under normal conditions, life and living are important; therefore, an oxymoron may emerge: an important but empty life—why important? Indeed, my life is important as the only life that offers me something that is everything I can be and have. Therefore, even an empty life is important from the first-person point of view: I have no hope for a better existence without life—or being alive. Optimally, life is meaningful and rich with varied and interesting content. I may wish my life would be like that. But a caveat applies. Life may be rich because it is full of accidents, hurts, and sorrows. Such a life is full of narrative content and is, ironically speaking, interesting. However, no one wants it. Reject it and leave it behind; do everything to change the future course of your life. A good life is rich, interesting, and desirable—it has a surplus of positive content.

What if one's life is too rich with good things? An overcomplicated and hurried living is suboptimal because it requires so much control and management to attain coherence. A good life is rich and interesting in moderation and entails a life with enough organized content—one must avoid the chaos that cannot be narrated. An overcrowded life and its problems fail to be interesting and enjoyable; they are interesting only ironically. One should live a moderate life of deliberation and planning because one enjoys the ride and does not want to spoil the fun. A messy life lacks enjoyment, just like a minimal life is

boring. People often report a desire to live simply, but this does not necessarily mean they want an uninteresting life. They want an opportunity to focus on what is essential in life, or truth, beauty, and love, which they see emerging only in simple contexts. This is prima facie a sensible idea. Others want a wild, risk-filled, and adventurous life—a meaningful life takes different guises. They want profound experiences, and others sheer variety. However, both may fail, one making life a mess, the other a mere bore.

Meaningful living shows coherence in its moderately rich desirable content. But people may live the life of others, finding inspiration in the narratives of their experiences, adventures, and achievements, both in real life and on TV. People not only live the life of others but also see their own life through their eyes. Their perceptions are therefore biased, desires copies, and self-evaluations simulations. We cannot avoid all such vicarious automatisms—we are social beings. Yet, to live a good life, a person must claim this life to be hers and sign it. She nurses and guards two positions: this life is mine, and I see and improve it personally. I sign my ownership claim, nurse its development, and guard the results because I do not want a life that is a copy to be copied by others. To avoid such an unduly vicarious existence, I should recognize its critical content as my own.[2]

Now, evil deeds and results are not my own. I fail to own them because I refuse to narrate my actions as they are and then claim and sign them—by thus confessing an evil life, I socially stigmatize myself, and nursing such a life makes my case worse; instead, I claim something else and pretend to nurse it. Evil hides and wears camouflage as it tries not to reveal itself. What does this mean?

An evil life, as such, is something the agent would not like to recognize: one cannot claim and declare it (Hallie 1982; Midgley 1985; Sklar 1985). A true, radical change from evil to good may fail, or the person cynically finds no need for it; instead, one seeks another strategy. An evil person pretends his life is good and wants to call it so. He wants his evil life to look good—a hypocritical wish and project. Within his evil existence lies a pretentiously laudable narrative when one explains a bad life favorably. The result is inconsistent because one has not crossed the narrative boundaries that should justify a positive evaluation of a fundamental transformation. Whatever he claims, his life remains an evil and alien existence. Yet he tends to be proud of what he is doing because he is good at it and thus demands admiration. He says his life is good, but his position remains essentially contested and alien to him when he conceitedly claims and nurses his life's illusory content. The hostile part of his audience will never agree with such a judgment because only the agents who face a similar rhetorical dilemma may agree with him. Such a controversy suggests that an evil life is never a coherent unity; one cannot claim, sign, and nurse it as such. Attempting to do so, one acts in bad faith. One cannot be consistently evil because one must camouflage some bad intentions and actions, and this leads to inconsistencies in the life plan.

## 2. Dreaming Life

Life is meaningful only if rich and interesting, and a rich life entails positive content we claim as our own. Where does it all come from? Things happen, and people enjoy good luck and suffer mishaps. A lottery win makes life meaningful, a bad car crash empty and meaningless. But people also create life's content. What is fundamental, acting or dreaming up events and actions? Literary authors such as Fernando Pessoa claim that being a dreamer is more meaningful than living an active life.[3] Dreaming trumps acting, says Pessoa. He claims that his inner world is so affluent because of its imagined variety of colorful visions and exciting events. One watches fiction emerging as a colorful circus of life or a bleak abyss.[4] Can dreaming form a fictional life rich with varied, interesting, and desirable content and thus create inviting imagery that is better than life itself and thus contributes to life's meaning as if from outside?

Alas, I cannot claim and sign a dream's content; I cannot step forward and declare it my own. I have nothing to show to others and no need to prove my ownership. My claim would be as imaginary as my dreams and my life's riches as fleeting and formless as

clouds in the sky. Everything is private, unreal, and fictional; therefore, life's meaning is also imaginary. Everything exists only in my imagination—if that stays intact: dreams are ephemeral and challenging to remember and describe.

Narrating dreams is to rewrite them. However, dreaming creates and narrates imaginary world fragments that can be extra rich because they are independent of what is. The dreamer says his life is rich because he can imagine anything. The dreamer prefers the circus. Suppose another agent masters his environment, has a good life plan, and knows why and how to act. He can live a meaningful life of action. Which life is richer and consequently more meaningful? I argue against the dreamer and support active life: in the end, dream content is private, transient, unpredictable, and inconclusive. A dreaming life is private and, as such, neither good nor evil. In my dreams, I cannot hurt anybody. My dreams do no good to anyone.

Suppose a person never communicates his thoughts through speaking or writing. He refrains from all actions. What you watch and contemplate in privacy may not be your own. I now explain this paradox of privacy and ownership.

Mental content belongs to me only if it is peculiar to me, but this presupposes agency. For example, I make a painting I sign and nurse as my own. I have created and recognized a unique object, which therefore is mine. Someone copies it and uses my signature—she steals it. An action puts the agent's stamp on some emerging facts of the world, and the agent consequently has a valid claim to them. He says these are mine. Now the two paintings are different, thus making possible two ownership claims: the two paintings must be different and, therefore, may have two separate owners. The painters cannot create two precisely similar objects; hence, we do not share a physical object. Leibniz's Law (of the identity of indiscernibles) says: No two substances are precisely alike. It follows that two different objects must have discernible properties that individuate them. Physical descriptions follow this law; what about the descriptions of mental objects?

Paradoxically, although mental objects are private and transient, others can copy and share my images, ideas, desires, and plans: their insubstantiality does not make them my own but, on the contrary, sharable. We are preparing to paint a nude model in class. We all see a naked body, and thus we imagine the same body. Of course, our images come from individual perspectives, but the image is still the same, namely the model's body. In this sense, we all paint the same object. Therefore, we all share the same mental image; as I argue, imaginary content is not my own or peculiar to me. Anyone can imagine this same thing, but no one can act like me: the paintings of the naked body are all different. The individuation and identification of private mental objects radically differ from epistemically shared physical objects. We painted the same mental and physical object, but all the paintings came out differently.

What about shared dreaming and dream objects? They are visual objects without real-world counterparts. Therefore, to communicate them, we use verbal descriptions. Private images do not figure in the social world except via non-individuating verbal reports. Are they, anyway, peculiarly mine? Can I claim and sign them? What if I dream of Marilyn Monroe? Myriads of people have entertained the same dream. Perhaps the details have been different so that my dream differs from yours.

Suppose my Marilyn is clothed and yours naked. Now our dreams are different. But millions of people have dreamed of a naked Marilyn, so your dream is not unique. Even if your image is rare, it can still be shared. Whatever details and differences you mention, the situation does not change; others can share the mental image. Hence you cannot sign and claim it. You cannot sign and claim a dream image in your dream world where objects are and remain stereotypes. I imagine Marilyn and someone has the same dream; you imagine naked Marilyn, and someone has the same dream, I imagine old, bald, and clothed Marilyn, etc. I can never individuate my dream objects based on their descriptions because they do not exist or are my own.[5] Mental objects are types, not individuals. Unlike my mental images, I can sign my physical objects, for instance, my paintings.

### 3. Sustainability and Transcendence

A good life is ethical, but an ethical life may not be good. Inactive living can be ethical, even if it is not good: you cannot do wrong if you live alone and do nothing. Let us discuss immanent and transcendent goals to understand what is at stake. Suppose I am inactive to the degree that I merely subsist. Call this sustainability, or the minimal definition of staying alive, if not living a life. Hermits in the wilderness have aimed at sustainable lifestyles.[6] However, the choice of a hermit's ascetic life requires critical reflection and a good reason. Let us assume that her reasons are secular: she dislikes company. But hermits also self-isolate for religious reasons. Their asceticism means preparing for the Upper Room to unite with their Creator. Such a goal is transcendent because it reaches beyond the experienced world, whose limits, however, one can only cross by special methods. The hermit does what she can to cross the border, thinking she can succeed only via an ascetic life.

Her life is now ethical in a minimalist sense, but is it a meaningful existence and a good life? From a secular point of view, she fails. The most obvious reasons are a trivial action repertoire, an unsociable lifestyle, and, most importantly, an illusory dream of life's purpose and goal. We should not recommend her lifestyle to others, but we leave her alone as she does not harm anyone. This is a secular reaction. Religiously, we may find her life admirable because of its sacred goal. In this case, we must develop a novel theory of the good life in which her purpose and goal play a vital role, and the role of meaningfulness is minimized. The current theory, however, says her life is meaningless; therefore, it is not good. We need not evaluate the goal to reach this conclusion. The hermit may agree that her life is meaningless, but she may add that this is a limited sacrifice she is willing to suffer for an infinite reward.[7] Some religious people, call them mystics, argue that silently contemplating the godhead constitutes the most meaningful existence—or the richest. Monks and nuns in monasteries believe that the dreary life of *ora et labora* is the best way to Heaven, which constitutes a sacred purpose and goal.

Cenobitic monks and nuns live cloistered lives hoping for the transcendent goal of salvation. Giorgio Agamben (2013) explains how to live under the principle of *ora et labora*, or pray and work, associated with the Benedictine Order. Superficially, "praying and working" sounds like a common-sense norm that may lead to a good life when understood secularly. But as Agamben explains, the case is more intriguing. The monks do not consider or deliberate about their life because the strict monastic rule they follow arranges and dictates every aspect of daily activities. This keeps them fully occupied to minimize their secret dreams and contemplative lives. They cannot flirt with any imaginable circus or contemplative abyss. The rule leaves them no space for desires, wishes, sentiments, longings, or temptations, as they have one motive only: to follow the rule and earn a place in Heaven—the monastic rule will take them there. And after following the rule long enough, they no longer say they want to follow it—they do so automatically. They have reached the stage where they want nothing and entertain no desires, yet they remain active in a simultaneously personal and automatic way. Moreover, they live in a well-organized and sustainable social environment.

The monks' life is repetitive and uninteresting, and it is not their own; therefore, it cannot be a good life from a secular perspective. They cannot claim, sign, and nurse the action results. Understood transcendentally, their failure is not obvious because of the otherworldly goal. Not all Christian theological schools agree that a life that denies the value of earthly living can be good, and the monk's life is indeed its negation. It resembles the dreaming life, except it does not leave room for the individual imaginary circus. On the contrary, the cenobitic life contains a perfectly conventional dream of salvation—which they call a mystery. For the monks, the good life is not here but there. The dreams of a secular person have no "there". Hence, one must try to live a good life "here".

### 4. The Logic of Meaning, Purpose, and Value

Action is essential for a meaningful life and living. By action, I mean human bodily activity with its reasons, results, and consequences. Let us discuss the results first. When I act, my performance connects to the facts of the world. I open a window, and the result is an opening window, and the consequences are an open window (proximal consequence) and a ventilated room (distal consequence). I cannot open the window without the window opening, although the expected consequences may not follow: they connect to action contingently. This scenario applies to success verbs, but process words are trickier: I push a rock that may or may not move, yet I push, and the rock gets pushed in both cases. If the window does not open, I have not opened it but tried to open it; when I push the rock, I push regardless of success. Without success, consequences may fail to emerge. However, I am active both when opening and pushing.

Acting supplies content, meaning, and a purpose to a good life—when the purpose supervenes upon the action. But the action results must be positively interesting, show variety in moderation, and be one's own. Moreover, the various actions must reveal something like a purpose-specific script, plan, program, or several such plans in unison. They must make sense when viewed together. Confused actions create random noise without meaning. Ideally, one acts according to a structured personal plan that supports life's purpose and, consequently, a good life. Of course, actions also come about independently of the plan, but then they fail to contribute to the purpose and meaning of life. Various incidents come and go without a trace, but serious errors and accidents may ruin everything. One's life may never recover.

The meaning and purpose of living are the two essential elements of the good life, and this suggests that purpose entails meaning, and meaning entails purpose. Although meaning and purpose are connected, one must not conflate them. I argue that the good life has its purpose that supervenes upon meaningful action. A good life is never without its purpose, understood in terms of consequences, yet purposes may occasionally be insignificant, hollow, and pointless. Nevertheless, a good life interconnects meaning and purpose. If all goes well, meaning and purpose together create the good life.

The purpose of life is a matter of the intended and anticipated consequences of actions. An agent acts, results follow, and consequences emerge. A painter paints, and the brush strokes make a painting as he intends. Why wield the brush if you settle with mere brush strokes and fail to finish the consequent picture? Meaningfulness entails a purpose. Sisyphus pushes a big stone up a hill that then rolls down.[8] He starts pushing again, *ad infinitum*. This is his life's purpose, yet such a life is meaningless because its characteristic actions are repetitious and intrinsically uninteresting. However, Sisyphus may see meaning in his divinely prescribed toil. He may imagine the subtle glory of his arduous work and its partially successful consequences: he succeeds in reaching the top. He succeeds and then fails—success should terminate activity, but not in a repetitious case. Yet, he is proud of his toil in his dreams, even if it cannot constitute his life's meaning. Regardless of its purpose, Sisyphean activity is not meaningful: the actions are not what they should be. Yet, Camus and his followers claim Sisyphus may narrate his life in terms of his purpose to make it a good life. "Sisyphus pauses, recognizes his suffering, accepts that is all there is, laughs at it, and thus achieves a self-consciousness worthy of his torment".[9] He is comforted and happy, but this is not to say that he leads a good life: Sisyphus's actions are meaningless, and thus his goal is also suspicious.[10]

As I argued, meaning entails purpose; therefore, no purpose entails no meaning. We need a clear picture of the purpose to understand the meaning—we cannot discuss them separately. Therefore, the purpose is a special thing that fits a meaningful life as a good life. Good goals and purposes presuppose a plan that sketches an overall target and the road there. Ideally, but only ideally, we need a plan that covers the minor plans making sense of the meaning-providing activities. Think again of the painter who paints but never finishes a canvas. He has no goal, so his painting activity fails to be meaningful. A finished painting should be a sub-goal; a thriving artistic life is the main goal and purpose. A good

life is a multi-layered affair, unlike Sisyphus's simple existence. The set of purposes makes action results meaningful *in toto*. A painter who wants to be a recognized artist (main plan and purpose) creates varied paintings (sub-goal) by painting (meaningful actions), and thus he leads a good life. If he is successful, his life is good. Suppose the painter destroys all his finished canvases and still aims at a successful artistic life—this is absurd in a Sisyphean sense.

Sisyphus represents a minimalist idea of the good life. What about maximal perfectionism? One can argue as follows. We must identify and realize life's ideally good positive goals and purposes; by denying or dismissing them, a person loses what is valuable in her life. Therefore, her life is not as good as it could be: this life can be good but never perfect (Buss 2006). Once she realizes this, she has a reason not to feel satisfied and must admit that her life is lacking. If a good life means a maximal condition, her life is not good. Life may look and feel good, but it could and should be significantly better, so she remains unsatisfied. If she aims at a good life, she must focus on what is the best available life. However, such a maximal notion of the good life may sound exaggerated. People may accept a more modest view that focuses on actions' positive and negative consequences—or on realistic achievements and avoidances—which form the good purpose behind all the sub-goals. If the purpose is good, all is well. It may seem unrealistic to think of the good life in a maximal perfectionist sense. A good life must be open to all; this requirement does not permit perfectionism.

Sometimes life's primary purpose allows us to define and recognize the agent: an enterprising painter becomes an established artist—and we recognize her as such. The famous British racing driver Graham Hill once called himself an artist: the car is his brush and the pavement his canvas. This sounds clever, but it does not change the fact that he was a racer. Racing was what he was doing, so it constituted the purpose of his life, in which winning was the goal. The main point is that life's purposes define the person; hence the purpose must entail value—a good life is an ethical life. What does this mean?

When an agent acts, she is responsible for adverse consequences, perhaps as collateral damage—this is unavoidable. She says she does not care about the damages, and hence she neglects a normative aspect of the consequences, proceeding as if they did not exist. Now, she must try to avoid bad consequences, which will be her life's minimal sub-project. But we must also ask about the value of the goals that make life good. As I defined it, meaningfulness is always good at the level of actions, unlike purposes. Is this to say that meaningfulness tolerates deficient purposes? How would this be possible? How can good meaning entail an evil purpose? I argued that life's meaning entails a purpose, and meaning is always good; yet good meaning would not entail an evil purpose—this is impossible. An evil purpose cannot supervene upon a meaningful set of actions. Good deeds cannot realize a bad goal. Think of parents who allow their child to eat all the candy he wants. At the action level, the parents do, prima facie, good, but the predictable consequences are bad. This is to say that, morally, their actions were bad.

Let us consider another example. A person is responsible for supplying a large household's daily food and drink. She conscientiously seeks the best, healthy ingredients and the finest drinks. Her life is rich, interesting, and her own—as such, she has a meaningful career. Her main goal is the welfare of the household. She cannot have a bad main goal that supervenes upon her daily activities. She may make mistakes and fail in many ways, but the associated harm is now irrelevant. We conclude that the meaningfulness of actions indeed guarantees a good goal and purpose. If her main goal is to enrich herself, she must act dishonestly, which robs her life of its meaningfulness—a meaningful life is never immoral. A good purpose supervenes upon meaningful actions and their results. They are laudable and make the agent justifiably proud—an emotion that celebrates the good life.

## 5. Against Happiness

Happiness is an obvious choice when discussing good goals and purposes in life. But it is a fuzzy notion; therefore, the question concerning good life and happiness is challenging.

At best, happiness is a family resemblance concept. The good life may be the same as the happy life. Or are they mutually independent? (Martin 2012). Of course, a good life tends to be happy, but a happy life need not be good. Good life tracks happiness—but a happy life ignores goodness. I do not need a good life to be personally happy, but if I want a good life, I may hope for happiness, too. Of course, happiness is, per se, a desirable state, yet a happy and evil life is possible. Evil people can be happy. Therefore, happiness is not a universalizable ethical term: my happiness may cause you pain—you are envious of my happiness, what makes me happy hurts you, or your unhappiness makes me happy. Because a happy and evil life is possible, we distinguish between a good and happy life: a good life is universalizable and ethical (Graham 2009; Brooks 2008). Happiness is a much broader category than the good life.

Let us specify the ideas of happiness. I skip traditional theories of happiness, for instance, the desire theory: happiness depends on the systematic satisfaction of desire.[11] A person is happy because of her satisfied desires. Aristotle in his *Nicomachean Ethics* (Aristotle 2011, 1095a15–22) explains that eudaimonia means doing and living well, or human flourishing, and this idea seems to correspond to my objectivist definition of the good life rather than the modern, subjective feel-good-theories of happiness. I cannot go deeper into the idea of eudaimonia here (Annas 1993).

I cover the following three ideas: happiness means (1) feeling good and being content with one's life, (2) personal hedonistic fulfillment, and (3) a virtuous life. The first is a folk-psychological notion because it cannot explain the birth of happiness. Be this as it may, happiness is essential in life, implying that one aims at a happy life as one's most significant goal. Assuming this is true, we live to be happy. To succeed, we act to make happiness an intended consequence of our actions, and happiness is the supreme purpose of life. We desire happiness. Two contingent possibilities exist. Either happiness just happens, or it is an achievement. As an unintended consequence of action, happiness depends on a lucky break so that happiness, as a desirable mental state, emerges as the contingent consequence of one's action results. Alternatively, I aim at happiness: I convince myself that marrying Alexandra makes me the happiest man on earth. These cases involve an indirect route to happiness: happiness is a contingent, unintended or intended consequence of an action that one naturally welcomes. I work hard and find myself happy, or I marry because I want to be happy. In these two cases, happiness happens if it happens.

A direct and non-contingent route is also possible. I play piano, and I am happy because I love to play. The agent need not hope for or want happiness because she knows what makes her happy. Creating music by playing the piano is a surefire happiness maker. I play to make myself happy because, for me, music is happiness.

We find two possibilities:

1.  An agent acts, and happiness follows because her acting has results, one of which is happiness. (Happiness makers are among action *results*.) An example: When I play the piano, I am happy.
2.  An agent acts and produces results with multiple consequences as goals, some of which are happiness makers. (Happiness makers are now among the *consequences*.) An example: When I play, an admiring audience makes me happy.

The quest for happiness may succeed without satisfying the requirements of a good life. People report that the most spartan and monotonous lifestyle is the best happiness maker (cf. (1) above), indicating a life with minimal meaning. In this case, life is not rich or interesting and often lacks a purpose. An idle and nondescript lifestyle may bring happiness but not a good life. Or life is rich and varied with a purpose that will cause unhappiness; suppose audiences never admire me, but I continue playing. I dedicate my life to helping others who remain ungrateful and even hostile. The consequences become unstable once the action exceeds a degree of complexity (cf. (2) above). This holds for intended and, of course, non-intended cases. Unlike the non-intended consequences, whose occurrences are ever-present and unlimited, no intended consequences may emerge.

Hedonism is a traditional theory of happiness: a person is happy because life is pleasant (Lampe 2014; Wolfsdorf 2013). Pleasure requires effort and invites maximization, which entails action. We maximize, but too much pleasure may turn into pain. Suppose we successfully regulate our pleasures. But to be happy, one may act egotistically and egoistically—which is not a good life. If one acts decently and in good faith, no happy enjoyment of life's pleasures may follow. Pleasures are tricky: they may require prior suffering, sometimes cause subsequent pain, and others are mixtures of pain and pleasure. Also, pain in suitable environments is pleasant, however paradoxical this may sound— think of SM practitioners in a club environment who enjoy torture. They enjoy receiving bodily pain and physically tormenting others. A happy life may be painful, yet life in pain is bad. One may refute this by saying the SM lifestyle may be both pleasant and good (Airaksinen 2017, 2018). But this makes a good life a non-universalizable norm; the argument illegitimately conflates happiness and the good life. People may also read an SM narrative as a horror story.

Hedonism comes in two guises: positive and negative. Epicurean hedonism is negative and thus free of the aforementioned troublesome aspects of happiness. Epicurus aims for a life of pain avoidance, understood as conflict, struggle, defeat, and disappointment. The aim is peace of mind or *ataraxia*, but we have already seen that it may oversimplify existence and thus fails as a meaningful life. The Sage recommends a shortcut to nowhere. Perhaps Epicurean hedonism can develop criteria of a meaningful life while making happiness its goal, thus making life good. However, I do not know how this argument combining omissions and actions would progress. The resulting connection between happiness and a good life should be so close that both terms have the same meaning. But a richer lifestyle may lead to too many struggles to allow cool and calm Epicurean *ataraxia* and happiness (Frischer 1982, chapters 1 and 2).

Historically, virtue as the source of happiness was the dominant theory for a long time.[12] Today the connection between virtue and happiness is no longer self-evident. Folk psychology does not recognize it, and philosophers fail to explain it. The idea is alien to modern ethicists, which is a pity (for an explanation, MacIntyre 2007).

The following reasoning is valid even if a virtuous life may not always be a good life: a good life is virtuous, and virtue is happiness; therefore, the good life is happy. But, as I mentioned above, evil happiness is possible, a view refuted by the virtue theory of happiness. Therefore, if the good life entails virtue, it entails a peculiar type of happiness. If this is true happiness, my theory is in trouble.

Why is a virtuous life a happy life? Virtuous action is a source of pride and allows no regrets, justified blame, or guilty feelings. Socrates claims this guarantees peace of mind and personal satisfaction in all circumstances. Virtuous action results make a person happy ((1) above), but now we end up with the familiar Socratic paradox: a virtuous person who suffers from harassment and unjust accusations is still happy. This may not happen because various disappointments may not allow a virtuous person's happiness. Therefore, unintended consequences may matter; yet, her peace of mind remains undisturbed, and she can still feel proud of herself and her actions. Her happiness remains intact because of this justified sense of pride—which is not a vice in this context (see Casey 1990). However, the Socratic paradox is real.[13] Schematically, the problem is as follows. A virtuous person is happy because of her virtue. At the same time, her virtue causes harm to her, which is prima facie a cause of unhappiness. To save the Socratic idea of happiness, we must say that virtuous happiness dominates the scene or that such suffering has nothing to do with happiness. Both alternatives look less than satisfactory.

Stanley Cunningham explains the happiness of a virtuous life. Virtue is a source of happiness when happiness is a "profound experience of flourishing, an expansiveness and tranquility of the soul", and, therefore, the supreme positive pleasure (Cunningham 2008, p. 261). The virtue theory collapses into positive and active hedonism, whose pleasures make the agent proud. Call this noble hedonism. Cunningham's virtuous happiness reflects (1) or (2). He says, "virtue is a source of happiness", which ambiguously refers to the results

and consequences of an action. Perhaps he first speaks of virtue in terms of (1) and then (2): virtuous action, as if immediately, produces happiness (1), but expansiveness and flourishing may presuppose (2)—unlike tranquility, which presupposes (1). Notice that different virtues work differently: courage and justice fall under (2), and the Aristotelian fundamental virtue of moderation falls under (1). The first two virtues, courage and justice, have diverse consequences, which make the virtuous agent proud and happy. The last, moderation, may have minimal consequences in the positive sense. Of course, omissions have consequences, but these are minimal compared to courage. Courage is an expansive virtue, while moderation is restrictive. We may live moderately because we control our actions' consequences, but this occurs indirectly: we primarily aim at modest action results. Moderation also works on expansive virtues to draw their circumstantial limits: too much courage means recklessness and too little cowardice, which are two vices. Virtues are like pleasures: beyond a limit, they turn into their opposites.

Is a virtuously happy life also a good life? As I have defined, a good life is much more than a virtuous life. A virtuous agent is proud to live a blameless, happy life. Yet, nobody can guarantee that his actions create no adverse, non-intended consequences—not even the perfectly virtuous Socrates. However, these need not make him unhappy, because he lacks responsibility for them. They are not his problem. But because of the bad consequentialist components, unlike Socratic virtue, the good life is not immune to unhappiness. A good life is a social existence with various kinds of social dependencies, which entail, for instance, vicarious suffering. Therefore, we cannot identify the good and virtuous life: a virtuous life is always happy, unlike a good life. And obviously, one can live a good life without being fully virtuous in the Socratic sense, or one can live a good life without showing a full set of virtues—it all depends on social circumstances. The good life does not entail virtue in the strictest sense of the term. A man of little courage can live a good life, and the same can be said of less-than-perfect practical wisdom, *phronesis*. *Sophrosyne* entails an ideal character but is not among the primary conditions of the good life. In a sense, virtue is a restrictive notion, unlike the good life. The good life may invite maximization, yet it cannot reach any ideal stage of perfection—unlike the idealized virtue.

Virtue leads to happiness mainly when we discuss (1). Then, we have virtue as a component of a meaningful life, which is happy because of its moral success and even perfection. But once we discuss (2), we may lose happiness, and then a good life is not happy; hence, virtue fails to include the good life. Therefore, the good life is happy and not happy, depending on where you look. And the good life need not be virtuous and happy; it is moderately ethical and perhaps unhappy. A virtuous life is the best life, but here we are interested only in the good life.

## 6. Authority, Autonomy, and Society

Consequences of actions occur and influence social situations in good and evil ways. Is it enough that my life's purpose does not harm anyone, or should I aim at the good of others in a positive sense? Omissive passivity may harm others through its egotism and moral indifference. Should a good life actively contribute to the good of others? A continuum of cases extends from omissive minimalism to saintly and heroic self-sacrifice (Urmson 1969). I fail to see how to specify the degree of the required social good that makes a plan and purpose optimal; instead, a certain degree of other-regarding goodness is necessary for a good life—this follows already from its definition. To make such a relativist approach sufficient, we may add minimal harm to the agent herself, minimal harm to others, social compassion, long-term viability, and cognitive comprehensibility, and t the plan must be one's own in the sense that one signs and nurses it. The agent should be proud of her life's plan and purpose.

Moreover, goals and purposes tend to be socially shared because they, for instance, require cooperation (Tuomela 2007). Purposes are not, therefore, unlike actions, self-owned in any straightforward sense. Or, perhaps, some social cooperation demands that the agent's actions are not her own but shared. Here we find another way of understanding

the meaning of life and bad projects: socially shared projects are not always under one's control. Actions and life's purpose may never be the agents' own in any demanding way. How one lives, wants to live, and evaluates one's decisions and goals depend on the social environment, education, other people, and social power relations. We must accept this, yet we can achieve a level of autonomy and be well-informed, critical, responsible, and active citizens. We can rise over and above our life's immediate environment and banal social relations to evaluate them. We may need to conform, but we also ask critical questions. We can and should avoid conformism, but how far does this take us? Also, our critical viewpoint depends on social influences; therefore, we must take the next step toward meta-criticism—or criticize criticism—and continue as if ad infinitum. We see our plans in a mirror reflected in another mirror, etc. Thus, freedom is relative: our goals fail to be entirely our own. Also, a person shares her efforts with others who claim the fruits of these actions. Yet, to live a good life, one must struggle toward autonomy and action ownership or mind the social virtues of independence, autonomy, responsibility, and accountability.

We all are socially lacking, which is unavoidable but not necessarily regrettable—especially if we know who we are, what we can be, and what we may try to achieve. Anyway, we need authorities to help us, especially if Joseph Raz is right with his "service conception of authority".

According to Raz, authority means the right to tell us what to do or believe. There are two types of authority, namely, practical authority, which tells us about what to do, and theoretical authority, which tells us about what to believe. Authorities mediate between the reasons for action and the subjects of the authority to whom those reasons apply. The directives of the authority have practical importance because they tell the subject how to act so that he does not need to directly consider at least some of the reasons that would bear on his acting in a particular circumstance (Gupta 2019, p. 1).

This type of authority is minimally restrictive and dangerous to individual autonomy—it may facilitate it. Authority, education, history, and culture form a background to the assertions of autonomy. Therefore, one should not regret the dangers of social life and aim at a hermit-like existence. On the contrary, we celebrate our social background and are proud of it. We cannot manage life, good or evil, without others. We need communities, friends, and relatives with whom we share social capital. We may recognize teachers and preachers as legitimate authorities, but we fight them not only because we want to be independent but also to get our social relations right. We are autonomous agents, and thus we wish to regulate our social connections—if we fail, they will overcome us. We are simultaneously restricted and free, so the struggle never ends, but it proves our autonomy anyway, and this is part of the good life. Our actions must be our own and their consequences good, so we are proud of them.

We depend on social contexts, authority, power, and other social relations: our actions influence others in ways they cannot fully control, but we also depend on them. My influence makes others' ownership claims shaky and vice versa. This arrangement is unfair if I claim full autonomy and independence from others when my actions still influence them. Let us universalize the arrangement: I am under the influence of others who may not claim full autonomy and independence because they are under my influence. Therefore, I accept the norm of mutual dependence within the bounds of our shared cultural context. The good life is social and, as such, open to moral considerations. Only a moral life can be good; at its core, we find meaningful action, sensible plans, purposes, and goals. Yet, ethical life is only an aspect of the good life: important and necessary actions may be irrelevant to ethics. And as I have shown, virtue does not always entail a good life: for instance, a virtuous existence may be less active than a good life. A cenobitic monk's virtuous life may not be good. Virtue tolerates omissions better than the good life. Also, the good life does not entail virtue, which is invariably happy.

**Funding:** This research received no external funding. Open access funding provided by Helsinki University Library.

**Institutional Review Board Statement:** Not applicable.

**Informed Consent Statement:** Not applicable.

**Data Availability Statement:** Not applicable.

**Conflicts of Interest:** The author declares no conflict of interest.

## Notes

1　I disagree with Wolf (2016). We can construe the idea of meaning in several ways.

2　Girard became permanently famous after he argued that all desires are copies, see Girard (1965, chp. 1; cf. my criticism in Airaksinen 2020).

3　See Pessoa (2013). Pessoa is not a passive dreamer because he writes voluminously and publishes. This looks inconsistent: a passive person actively writes and publishes. To avoid this accusation, in his book Pessoa narrates the inner life of a fictional dreamer. In this way, he tells a coherent story. Also, Rousseau (1992).

4　This is a romantic attitude, as explained by Berlin (1999, chp. 5).

5　I accept Wittgenstein's argument against private language, which supports my view of mental objects: "The words of this language are to refer to what only the speaker can know—to his immediate private sensations. So another person cannot understand the language" (Wittgenstein 2009, § 243; see Candlish and Wrisley 2019).

6　Westermarck (1939) is a forgotten classic in the field, see chp. 9.

7　This is related to Pascal's Wager, see Rescher (1985).

8　David Wiggins' (1988) work has influenced me; also Ellis (2011).

9　Vrana (2022). I take up this example because Sisyphus is such a popular test case.

10　Camus (1991) concludes his essay on Sisyphus by calling him happy. This is an absurd statement, but Camus aims at absurdities, which is not absurd. Camus's account makes us wonder what kind of non-standard and exceptional ways of meaning construction a person may find and ask how we should evaluate and criticize them.

11　See Airaksinen (2016). In their sprawling treatise on desire, Arpaly and Schroeder (2014) fail to discuss happiness, yet the reason why we work to satisfy our desires is our expected happiness.

12　See, for example, Cunningham (2008), Parts III and V.—Virtue and ethics are not synonymous. They represent two approaches to how to live a blameless life.

13　Weiss (2006); this study has generated discussions of the Socratic paradoxes.

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
