# Peer review of "Good Life without Happiness"

_humanities, doi:10.3390/h11060155_

Round 1
Reviewer 1 Report
SUMMARY
The author’s main objective is to define what a good life is and to establish how it is different from a virtuous life and happiness.
He argues against the idea that a passive life could be a good life.
He also discusses various theories about happiness showing that a good life may not always be a happy one, although it tends to be so. The author says: “Good life is universalizable and ethical. Happiness is a much broader category than the good life” (319-320).
His most original assertion is that a good life is not the same as a virtuous life.
He says: “Today the connection between virtue and happiness is no longer self-evident [..]The idea is alien to modern ethicists, which is a pity.”(385) Suggestion for author: see Alasdair MacIntyre.
The author continues: “Virtue may be the only chance if we want to save the connection between happiness and a good life” (386-387).
But then, the author’s conclusion seems to be contradictory: “the good life need not be virtuous and happy; it is moderately ethical and perhaps unhappy. A virtuous life is the best life, but here we are interested only in the good life” (447-449).
Suggestion for the author: Clarify this apparent contradiction.
GENERAL CONCEPT COMMENTS
1. The manuscript is clear, relevant for philosophical and social fields.
2. It is presented in a well-structured manner.
3. The cited references are relevant.
4. The paper will attract a wide readership.
Author Response
1) Ref1 says:
He says: “Today the connection between virtue and happiness is no longer self-evident [..]The idea is alien to modern ethicists, which is a pity.”(385) Suggestion for author: see Alasdair MacIntyre.
My answer:
I have added a reference to MacIntyre.
Ref1 writes: The author continues: “Virtue may be the only chance if we want to save the connection between happiness and a good life” (386-387).
But then, the author’s conclusion seems to be contradictory: “the good life need not be virtuous and happy; it is moderately ethical and perhaps unhappy. A virtuous life is the best life, but here we are interested only in the good life” (447-449).
My answer: Perhaps this is not a contradiction, but I have anyway corrected the text,
as follows: "Virtue may promise to connect happiness to the good life, but as I argue below, this does not happen."
Reviewer 2 Report
This is an interesting and enjoyable paper, but it could benefit from some revisions. I will try to indicate what these should be below.
The paper’s first paragraph is mysterious to me. The author says that an empty life will be tedious and disappointing “even if life were important from the person’s first-person perspective” (line 23). Does this mean that it matters whether someone’s life is tedious, disappointing, etc. to others? Surely the meaning of life does not lie in entertaining other people. Or is the idea rather that one’s life could be tedious, etc. to oneself despite also seeming important? If so, does this matter? Would I rather live an important life or an interesting one? I’m not sure. In short, it is not clear to me what the author is saying here or why s/he is saying it. Is this all meant to be obviously true, or is it a statement of a controversial thesis that will be argued for in the rest of the paper?
In the next paragraph the author says that “even an empty life is important from the first-person point of view” (lines 30-31). Is this always true? One question: what does it mean for a life to seem important? Does it mean simply that one’s death would be a very bad thing, or does it imply that the life in question is achieving something significant? Another question: don’t some people feel that their lives are both empty and unimportant (some suicidal people, for instance)?
In lines 32-33 of the same page the author says that “Optimally, life is meaningful and rich with varied and interesting content.” This seems debatable. A life like that sounds good to me, but couldn’t wanting varied and interesting content seem a bit shallow? Do very religious or very ethical people care much about this kind of content?
On p. 2 lines 48-49 the author seems to assume that we are talking about subjective or first-person point of view meaning. If so, I think this could be made clearer.
On lines 66-67 of p. 2 the author claims that one cannot claim and declare an evil life. Is this really true? Don’t some people pride themselves on being evil, or at least “evil”? There are many songs in which the singer proudly claims to be evil, and members of biker gangs, for instance, often seem to want to be seen as evil.
The author might be right that an evil life is never a coherent unity (p. 2 line 79), but I would like to see more argument in support of this claim.
On p. 3 line 109 the author says that since the content of a dream can be anything, it is nothing. I think this needs more explanation.
In line 110 of p. 3 s/he implies that what is private (e.g., dreams) cannot be good or evil. But can’t a dream be sadistic or magnanimous?
In line 112 of p. 3 the author talks about mystics together with dreamers, but mystics are not really discussed until the next section, which perhaps should be mentioned here.
On p. 4 line 151 the author asserts that mental objects are types, not individuals. Is this just generally accepted? Would it be worth arguing for this claim?
On p. 5 lines 217-218 the author says that a person’s actions should “only loosely and tentatively reveal something like a […] plan.” Why “only loosely”?
On p. 7 line 292 the author says that “meaning is always good.” Is this meant to be true by definition? If so, it might be worth saying so. I would think that someone’s life might have a very bad or evil meaning. I would not call such a life meaningful (without qualification), but if someone asked about, say, the meaning of Hitler’s life, I would be more inclined to say that it was very bad than to say that it simply had none.
Author Response
PLEASE, SEE THE ATTACHMENT.
